# Subverting the Canon: Novel Cancer-Promoting Functions and Mechanisms for snoRNAs

**DOI:** 10.3390/ijms25052923

**Published:** 2024-03-02

**Authors:** Matthew Huo, Sudhir Kumar Rai, Ken Nakatsu, Youping Deng, Mayumi Jijiwa

**Affiliations:** 1Krieger School of Arts and Sciences, Johns Hopkins University, Baltimore, MD 21218, USA; mhuo3@jh.edu; 2Department of Quantitative Health Sciences, John A. Burns School of Medicine, University of Hawaii, Honolulu, HI 96813, USA; raisk21@hawaii.edu (S.K.R.); knakats@emory.edu (K.N.); 3Emory College of Arts and Sciences, Emory University, Atlanta, GA 30322, USA

**Keywords:** snoRNA, ribosome, nucleolus, rRNA, SNHG, cancer

## Abstract

Small nucleolar RNAs (snoRNAs) constitute a class of intron-derived non-coding RNAs ranging from 60 to 300 nucleotides. Canonically localized in the nucleolus, snoRNAs play a pivotal role in RNA modifications and pre-ribosomal RNA processing. Based on the types of modifications they involve, such as methylation and pseudouridylation, they are classified into two main families—box C/D and H/ACA snoRNAs. Recent investigations have revealed the unconventional synthesis and biogenesis strategies of snoRNAs, indicating their more profound roles in pathogenesis than previously envisioned. This review consolidates recent discoveries surrounding snoRNAs and provides insights into their mechanistic roles in cancer. It explores the intricate interactions of snoRNAs within signaling pathways and speculates on potential therapeutic solutions emerging from snoRNA research. In addition, it presents recent findings on the long non-coding small nucleolar RNA host gene (lncSNHG), a subset of long non-coding RNAs (lncRNAs), which are the transcripts of parental SNHGs that generate snoRNA. The nucleolus, the functional epicenter of snoRNAs, is also discussed. Through a deconstruction of the pathways driving snoRNA-induced oncogenesis, this review aims to serve as a roadmap to guide future research in the nuanced field of snoRNA–cancer interactions and inspire potential snoRNA-related cancer therapies.

## 1. Introduction

The intracellular orchestra of protein synthesis unfolds through a delicate interplay involving small nucleolar RNAs (snoRNAs), small nucleolar RNA host genes (SNHGs), ribosomal RNAs (rRNAs), the nucleolus, and ribosomes. SnoRNAs, derived from specific genomic loci such as SNHGs, intricately direct the modification and processing of rRNAs within the nucleolus. Mature rRNA forms the architectural backbone of ribosomes, the molecular machines that translate genetic information into functional proteins. In recent years, snoRNAs and their associated factors have attracted growing attention. Numerous publications have revealed that these elements are individually or collectively implicated in various human diseases, including cancer, to a greater extent than previously believed.

SnoRNAs are conserved noncoding RNAs with a well-known canonical role in the biogenesis of rRNAs. In humans, snoRNAs, which vary in size from 60 to 300 nt, typically accumulate in the nucleolus and are primarily derived from SNHGs [1,2]. SnoRNA biogenesis in eukaryotic cells, tightly regulated by RNA polymerase II, is involved in processing the 47S preribosomal RNA transcript with the spliceosome complex [1,3,4]. In this process, snoRNAs act as stabilizing components by forming ribonucleoprotein complexes called small nucleolar ribonucleoproteins (snoRNPs) [1,2]. There are two major groups of snoRNAs, box C/D and box H/ACA snoRNAs, which direct methyltransferase and pseudouridine synthase enzymes to the modification site via complementary base pairing with the target rRNA [5,6,7,8].

In addition to these canonical functions in the nucleolus, snoRNAs are involved in non-canonical mechanisms, such as the biogenesis of piwi-interacting RNA (piRNA) and snoRNA-derived fragments (sdRNA), the derivation of certain microRNAs (miRNAs), tRNA methylation, and rRNA acetylation [9]. Moreover, snoRNAs turned out to play diverse roles in RNA-related processes, regulating splicing and chromatin accessibility [9]. All of these epigenetic processes play a crucial role in the proper execution of cellular bioactivities.

While many snoRNAs are associated with the modification of specific RNA targets, some are classified as orphan snoRNAs, which have no known targets. Han et al. systematically profiled chromatin-associated snoRNAs (casnoRNAs) in mammalian cells and identified a subgroup of orphan casnoRNAs that responded to DNA damage stress [10]. In addition, snoRNA-derived small RNAs have been reported to regulate gene expression in various ways, such as by functioning like miRNAs. This leads to their involvement in the development of various pathological conditions, including cancer [11,12,13,14,15,16,17].

Most snoRNAs are derived from the introns of SNHGs (Figure 1(2): snoRNA parent transcript splicing). SNHGs, which can be protein-coding or noncoding genes, produce primary transcripts in the nucleus. These transcripts undergo splicing, with exons directed to cytoplasmic functions as either protein-coding mRNAs or noncoding RNAs (long noncoding RNA SNHGs; lncSNHGs) (Figure 1(3b): Spliced lncSNHG/mRNA) [9,18,19]. Introns yield mature snoRNAs, which then localize to the nucleolus and Cajal bodies (Figure 1(3a): Post-transcription processing) [20,21,22,23]. LncSNHGs have emerged as a class of transcriptional regulators, playing pivotal roles in cancer development as both oncogenes and tumor suppressors [2,18,20,23,24,25,26,27]. The upregulation or downregulation of lncSNHGs in cancer is correlated with increased proliferation, invasion, and metastasis, and the involved molecular mechanisms have been identified in several cancers [20,23,28,29].

The nucleolus and the ribosome are intracellular organelles that were identified decades ago [30,31]. Interestingly, with the advancement in technology, these well-established organelles are receiving new attention. The nucleolus is a subnuclear membraneless organelle that is the epicenter for rRNA processing and ribosome biogenesis [32,33]. It comprises three distinct regions: the fibrillar center (FC), the dense fibrillar component (DFC), and the granular component (GC). Evidence is now mounting that a phenomenon known as liquid–liquid phase separation (LLPS) underlies the formation of membraneless “liquid droplet” compartments, such as the nucleolus, in cells [30,34,35,36,37,38]. Ide et al. revealed that mutations in RNA polymerase I induced structural changes in nucleoli and abnormalities in ribosome synthesis, leading to human hereditary diseases [39].

The ribosome is the hub of protein synthesis, where polypeptide chains are synthesized by translating the genetic information encoded in mRNA. When abnormalities occur in ribosome biogenesis, normal translation is compromised. This leads to the disruption of various cellular systems. Human diseases that are thought to be caused by the failure of normal ribosome synthesis are being discovered and are collectively referred to as “ribosomopathies” [40,41,42].

This review presents the latest insights into snoRNAs and their cooperative mechanisms. Subsequently, we introduce strategies that have been reported and incorporate these factors into various cancer therapeutic approaches.

## 2. Characteristics and General Functions of snoRNA

### 2.1. Classification of snoRNAs

#### 2.1.1. C/D Box

C/D box snoRNAs, collectively known as SNORDs, are characterized by their kink-turn (also called the stem–bulge–stem) structure [8]. As their name suggests, C/D box snoRNAs contain C and D boxes, which are conserved sequence motifs present in canonical C/D box snoRNAs (Figure 2a) [43]. C boxes consist of the sequence motif RUGAUGA (R standing in for a purine base), while D boxes represent the sequence CUGA [44]. In addition, C/D box snoRNAs contain a less conserved C′ and D′ box located upstream of the D box and downstream of the C box, respectively (Figure 2a), which have identical nucleotides to their counterparts [45]. Upstream of the D and D′ boxes are antisense elements known as “guide regions”, which are complementary to target RNAs and allow for the precise methylation of targets (Figure 2a) [44]. Using their scaffold-like structure, conventional SNORDS attract and position partner proteins, canonically including the (human/yeast) fibrillarin (FBL)/Nop1p, SNU13 (NHP2L1)/Snu13p, NOP58/Nop58p, and NOP56/Nop56p to form a small nucleolar ribonucleoprotein (snoRNP) (Figure 3a) [44,46]. Partner proteins appeared to interact reciprocally in the accumulation of snoRNAs as well [47]. snoRNP complexes facilitate the canonical function of SNORDs: transferring a methyl group to 2′ oxygen on a ribose molecule in a process known as 2′-O-methylation (Figure 3a) [44]. These proteins give the snoRNP functionality and prevent its exonuclease-mediated degradation while ensuring proper localization within the nucleolus [15]. Some C/D box snoRNAs were found to facilitate the acetylation of the critical 18S structural rRNA in eukaryotes, allowing acetyltransferases to catalyze the formation of the ac4C modification [43].

#### 2.1.2. H/ACA Box

H/ACA box snoRNAs, known as SNORAs, also contain the kink-turn structure. However, they contain two additional hairpin structures and different sequence motifs and have corresponding differences in functions (Figure 2b) [43]. H/ACA box snoRNAs’ canonical function is to facilitate the RNA-dependent pseudouridylation of their target RNAs, which primarily consist of pre-rRNAs [48]. Pseudouridine is the most common isomer of uridine, and the pseudouridylation of rRNAs is critical for proper ribosomal function in eukaryotes [48]. These RNAs’ hinge (H) and ACA boxes, which consist of the sequence motifs ANANNA and ACA, respectively, are located downstream of their kink-turn structures (Figure 2b) [44]. Within the internal loops of an H/ACA box snoRNA are pseudouridylation pockets containing antisense elements, which bind to target RNA via complementary base pairing and allow for the precise pseudouridylation of targets (Figure 2b) [48]. Conventional H/ACA snoRNAs form snoRNP complexes with NHP2/Nhp2p, NOP10/Nop10p, GAR1/Gar1p, and the pseudouridine synthase dyskerin (DKC1)/Cbf5p (Figure 3b) [44,46].

#### 2.1.3. scaRNAs

Lastly, the small Cajal body RNA (scaRNA) subfamily consists of snoRNAs with the structural characteristics of the C/D box and H/ACA RNAs and occasionally both [43]. Although scaRNAs can fall under the C/D-H/ACA classification, their unique localization in nuclear Cajal bodies distinguishes them from other families. scaRNAs also contain distinct sequence motifs, including the CAB box and GU-repeat elements, which are necessary for the proper localization of scaRNAs in Cajal bodies [49]. Within Cajal bodies, scaRNAs guide 2′-O-methylation, pseudouridylation, and other modifications to process various RNA species, similarly to their C/D and H/ACA box counterparts [44].

### 2.2. Canonical snoRNA/snoRNP Biogenesis

The common pathways of snoRNA biogenesis are well established in the literature. As the majority of studies that characterize this elaborate process were conducted on yeast samples, the processes that will be discussed occurred in yeast unless otherwise specified, and the yeast homologs of proteins will be named, though the process is assumed to be comparable to that of humans due to the high degree of conservation between human and yeast intron-derived snoRNAs [15,50]. Still, the use of this model species presents limitations to the current understandings of snoRNA biogenesis in humans, which additional studies are needed to clarify.

Most human snoRNAs are embedded in the introns of noncoding snoRNA host genes (SNHGs). However, some snoRNAs are found within the introns of protein-coding genes (Figure 1(2): snoRNA parent transcript splicing) [51]. snoRNAs can be transcribed in a mono- or polycistronic manner, though polycistronic clustering occurs infrequently in humans [51]. Human snoRNA-embedded primary transcripts are transcribed by RNA polymerase II [51]. Nop1p binds to snoRNAs and associates with Rnt1p, an endonuclease which binds to the external stem structural motif. Rnt1p and Nop1p then cooperatively cleave snoRNAs confined in mono- and polycistronic sequences from the primary transcript into intron lariats, which are enzymatically debranched and linearized and trimmed by exoribonucleases packaged in the nuclear exosome in order to become functional transcripts (Figure 1(2): snoRNA parent transcript splicing) [52,53,54,55]. snoRNAs undergo slightly different processes at the 5′ and 3′ ends, with Nop1p and Rnt1p targeting sites present on the 5′ end and removing the 5′ cap, while on the 3′ end, endonucleolytic cleavages map entry points for exonucleases to continue processing the snoRNA (Figure 1(3a): Post-transcription processing) [52,56].

Simultaneously, to protect the molecule from immediate post-transcriptional exonucleolytic degradation, snoRNP proteins assemble cotranscriptionally on nascent snoRNA transcripts, forming precursors to the canonical snoRNP [57,58]. For H/ACA snoRNAs, Naf1p, a H/ACA snoRNP assembly factor, recruits Cbf5p and Nhp2p, the catalytic subunit of H/ACA snoRNPs—the yeast homolog of dyskerin [59]—and a H/ACA snoRNP core protein, respectively, which associate with the C-terminus of the RNA polymerase II (RPB1) subunit [58] at the 3′ end of the snoRNA [59], which recruits several other RNA-binding proteins, including Nop10, a canonical snoRNP core protein (Figure 1(1): Cotranscriptional snoRNP assembly) [57,59]. Gar1p is posttranscriptionally recruited, indicating that it is not directly involved in snoRNA synthesis [57]. SHQ1 may also play a vital role as a chaperone molecule for Cbf5p [60]. For human C/D box snoRNAs, the mature snoRNP complex is sequentially assembled, beginning with Snu13, which recognizes structural motifs of the C/D and C′/D′ boxes, forming a precursor complex which is recognized by Nop56 and Nop58, whose N-termini recruit Fibrillarin (Figure 1(1): Cotranscriptional snoRNP assembly) [58]. Both assembly pathways likely largely involve the Hsp90 and R2TP complex chaperone molecules, which stabilize RNP core proteins and coordinate a large array of assembly factors [61].

snoRNAs generally possess two different kinds of 5′ cap modifications, including the 7-methylguanosine cap and monomethylphosphate cap [62]. RNA polymerase II-transcribed molecules generally undergo trimethylation [62]. However, the intron-derived snoRNAs most common in humans tend to receive no cap modifications [63]. These canonical snoRNAs lack the traditional structural features, such as the 5′ monomethylguanosine cap and poly(A) tail, present on mRNAs that undergo nuclear export [64], which was used to explain their localization in the nucleus, though recent reports of noncanonical snoRNA activity in extranuclear cellular localizations easily disprove this notion [64]. snoRNAs may escape the nucleus via stress-regulated transport/shuttling proteins, including NXF3 and DBR1 [55,65].

One peculiar feature of snoRNA expression—its apparent disconnection with its host gene’s expression—may be explained based on the mechanisms by which snoRNA processing occurs [66,67,68]. snoRNA transcription is duly initiated with promoters containing canonical pyrimidine-anchored transcription start sites in proximal or overlapping configurations with noncanonical 5′TOP initiators to form a hybridized dual promoter [69]. In addition, endonucleolytic cleavage serves to excise snoRNAs from their host sequences but also uncouples the expression of the two, likely as a result of nonsense-mediated decay [70]. Thus, the unrelated expression of snoRNAs and their host genes is speculated to be a byproduct of the normal processes of snoRNA synthesis.

### 2.3. snoRNAs’ Canonical Roles in rRNA Processing

snoRNAs are known to play a critical role in RNA processing, primarily targeting ribosomal (rRNAs) and small nuclear RNAs (snRNAs) [71]. In spliceosomes’ snRNAs, modified bases are concentrated in regions associated with pre-mRNA splicing and are functionally critical, regulating splice-site recognition and the formation of small nuclear ribonucleoprotein (snRNP) complexes that form the spliceosome, among other characteristics that are necessary for proper splicing [72]. Since most snoRNAs included in this review had some canonical interaction with rRNAs, we will highlight this aspect of snoRNAs’ canonical activity. To briefly summarize the complicated involvement of snoRNAs in rRNA processing, in human nucleoli, a cohort of snoRNAs are involved, with many ribosomal proteins and pre-ribosomal factors, in the endonucleolytic cleavage of pre-rRNA transcripts surrounding internal transcribed spacers (ITS) during the initial processing of the full-length 47S pre-rRNA precursor transcript to isolate ribosomal subunit pre-rRNAs for further processing (Figure 3c) [73]. Using their antisense complementarity, these snoRNAs, assembled into snoRNPs, base pair with pre-rRNAs at docking sites throughout the precursor transcript and participate in the regulation of the transcript’s structure [73]. During this stage, snoRNPs can prevent premature folding at their binding sites, while other snoRNPs concomitantly induce modifications to trigger structural changes along the precursor transcript; the snoRNPs simultaneously prevent premature misfolding at those sites and alter the pre-rRNA’s structure at other locations, while endonucleases facilitate the cleavage of the precursor transcript, ultimately ensuring the excision of structurally correct pre-rRNA transcripts [73,74,75,76,77,78]. Once this process is completed, RNA helicases and other RNA-binding proteins trigger the release of the snoRNPs [73,78]. snoRNPs mediate modifications, which alter functional groups of nucleotides and, consequently, intramolecular interactions within the precursor transcript that are putatively responsible for its three-dimensional folding [79]. However, the fact that some snoRNPs specifically bind to pre-rRNA transcripts but do not mediate any modification and are necessary for the production of functional pre-rRNA transcripts points to the existence of snoRNPs that instead direct nucleases to sites for cleavage [80]. pre-rRNAs also undergo exonucleolytic trimming to remove external transcribed spacers via complex sets of processes, which are different for each type of pre-rRNA being processed [73,77]. For the processing of 18S pre-rRNA, several snoRNAs and the spliceosomal subunit processome, comprised of many subcomplexes, including U3 snoRNP, which is necessary for processome formation, are essential for this trimming [74,77].

### 2.4. Noncanonical snoRNA Functions

snoRNAs are a versatile RNA species involved in a range of cellular processes. Recent research has largely identified noncanonical functions for snoRNAs, which define a vast range of unexplored snoRNA activity.

Some snoRNA transcripts are further processed (Figure 3f) by the microprocessor complex and endoribonucleases, including DICER or SLICER, into snoRNA-derived fragments (sdRNA), which are recruited into post-transcriptional silencing complexes and function similarly to miRNA (Figure 3h) [11,81,82]. Both H/ACA snoRNA and C/D snoRNA can produce miRNA or sdRNA. Products derived from H/ACA snoRNAs are mostly 20–24 nt long and originate from the 3′ end, while those from C/D snoRNAs exhibit a bimodal size distribution at around 17–19 nt and >27 nt, predominantly originating from the 5′ end [12]. Similar to conventional miRNAs, many snoRNA-derived miRNAs and sdRNAs target mRNA at the 3′UTR and some at the 5′UTR or coding regions [11]. “Orphan” snoRNAs, which show no specific RNA base-pairing complementarity, are speculated to comprise many sdRNAs [16]. Several snoRNA-derived piRNAs have been observed to be associated with PIWI proteins and participate in epigenetic regulation in mammalian somatic cells, directing the exchange of the H3K4me3 and H3K27me3 histone modifications on gene promoter regions, binding to mRNAs, and recruiting TRAMP protein complexes, which are involved in nucleolytic RNA degradation (Figure 3g) [14,83]. A few C/D box snoRNAs have also been identified to participate in pre-mRNA alternative splicing. The snoRNA-mediated 2′O methylation of an intron substrate’s branch point adenosine blocks the critical transesterification reaction between its 2′ hydroxyl group and the phosphodiester bond of the guanosine and neighboring base at the intron’s 5′ splice site (Figure 3d) [84]. When splicing normally occurs, the phosphodiester bond is broken, resulting in a new bond between the adenosine and guanosine [84]. The new bond forms the loop structure of the intron lariat and is necessary for proper splicing and the mature mRNA transcript’s appropriate expression [84,85]. However, when the adenosine 2′ hydroxyl is blocked by a methyl group, exon inclusion and exclusion can occur, leading to the synthesis of alternate proteins from the same transcript (Figure 3d). SNORD13 was found to be associated with the NAT10 acetyltransferase and guide the transfer of the N4-acetylcytidine (ac4C) modification to 18S rRNA in humans by complementary base pairing—in a manner identical to its expected canonical function (Figure 3e) [86]. AluACA snoRNAs, which possess an altered version of the double hairpin structure of H/ACA snoRNAs, also lack functional binding sites with canonical snoRNP proteins and are not functionally well characterized [87]. Most noncanonical snoRNAs covered in the recent literature, however, are snoRNAs that fall under normal C/D and H/ACA box classifications and canonically guide the modification of RNAs in addition to noncanonically mediating oncogenesis. Thus, these transcripts will be the focus of this review.

### 2.5. Small Nucleolar RNA Host Gene (SNHG)

The SNHGs are a group of genes that can be processed into snoRNAs and lncSNHGs [9,18]. The HUGO Gene Nomenclature Committee has published more than 30 SNHGs, including *SNHG1* to *SNHG33*, *GAS5*, and *ZFAS1* [88]. These SNHGs are first transcribed into primary SNHG transcripts and then further spliced into exons and introns (Figure 1) [20,21,22,23]. The exons are then re-spliced and translocated to the cytoplasm to function as protein-coding mRNA or noncoding RNA, i.e., lncSHNGs. Intronic sequences are further processed into mature snoRNA [9,18].

LncSNHGs are a type of lncRNA. Recently, they have attracted attention as an emerging gene transcription regulator that functions as either an oncogene or tumor suppressor [2,18,20,23,24,25,26,27]. Their regulatory roles in tumorigenesis involve various aspects of biogenesis, such as acting as miRNA sponges, inhibiting protein ubiquitination, and enhancing DNA methylation [9,18,89]. Their aberrant expression also influences the abnormal behavior of cancer cells, including epithelial–mesenchymal transitions (EMT), cell cycle progression, proliferation, invasion, and the evasion of apoptosis [28,90,91,92,93,94].

Overall, SNHGs and their lncSNHGs may play crucial roles in the future of cancer therapy. Consequently, elucidating the molecular mechanisms underlying the correlational links between these RNAs, cancer, and immune responses is necessary to contribute to the development of novel therapeutic approaches.

### 2.6. Nucleolus

The nucleolus, a vital membraneless organelle within the nucleus, is a central hub for rRNA processing and ribosome biogenesis [32,33]. Dysregulated nucleolar function leads to abnormalities in ribosomal biogenesis, resulting in various ribosomopathies, including cancer [95,96,97]. Structurally, the nucleolus consists of three discrete regions, namely, the fibrillar center (FC), dense fibrillar component (DFC), and granular component (GC) [98]. Recent advancements in high-resolution live-cell microscopy have allowed for the identification of 12 proteins, including unhealthy ribosome biogenesis 1 (URB1), enriched toward the periphery of the DFC, named the peripheral dense fibrillar compartment (PDFC) [99]. URB1, a static nucleolar protein, is crucial in anchoring and folding the 3′ end of pre-rRNA. This ensures the recognition of U8 small nucleolar RNA and the subsequent removal of the 3′ external transcribed spacer (ETS). The depletion of URB1 disrupts the DFC, causing aberrant pre-rRNA movement and altered conformation, activating exosome-dependent nucleolar surveillance with downstream effects on rRNA production and embryonic development.

Another nucleolar protein, polyglutamine-binding protein 5 (PQBP5 or NOL10), binds to polyglutamine tract sequences and constitutes the skeletal structure of the nucleolus [100]. This protein remains stable under stress conditions, anchoring other nucleolar proteins during osmotic stress and maintaining the nucleolar structure. The functional depletion of PQBP5/NOL10, as seen in polyglutamine disease proteins, leads to pathological nucleolar deformities or disappearance.

Another notable development in the study of the nucleolus was made in the field of liquid–liquid phase separation (LLPS). This membraneless organelle, the nucleolus, is sequestered from the nucleus by liquid droplet formation through LLPS. For the nucleolus and other sub-nuclear organelles, the formation and regulation of LLPS are closely associated with oncogenesis, tumor progression, and metastasis [101,102]. Ide et al. used single-molecule tracking to show that RNA polymerase I (Pol I) and chromatin-bound upstream binding factor (UBF) undergo transcription suppression through phase separation [39]. Active Pol I forms small clusters in the FC, restricting rDNA chromatin. The inhibition of transcription causes Pol I to disassociate from rDNA, becoming liquid-like in the nucleolar cap. A Pol I mutant linked to a craniofacial disorder competes with wild-type Pol I, transforming the FC into a cap and inhibiting transcription. The cap droplet excludes an initiation factor, ensuring effective silencing. This reveals a mechanism of rRNA transcription suppression via Pol I-mediated phase separation within the nucleolus.

Condensates induced by transcription inhibition (CITIs) in the nucleolus drastically alter the spatial organization of the genome. CITIs are formed by the splicing factor proline- and glutamine-rich (SFPQ) protein, the non-POU domain-containing octamer-binding protein (NONO), fused in sarcoma (FUS), and TATA-Box-binding protein-associated factor 15 (TAF15) in nucleoli upon the inhibition of RNA polymerase II (RNAPII). Yasuhara et al. found that the SFPQ protein and NONO undergo rapid LLPS in nucleoli upon RNAPII inhibition, resulting in the formation of CITIs [103]. The localization of active chromatin to CITIs increases the illegitimate fusions of DNA double-strand breaks (DSBs) in active genes, promoting the formation of fusion oncogenes. It has been suggested that proper RNAPII transcription and rRNA processing are essential for preventing the LLPS of SFPQ/NONO on rRNA.

## 3. snoRNA and Oncogenesis

Although advancing early screening and treatment options have generally improved long-term cancer survival rates, the cellular interactions through which snoRNAs are implicated in the disease have not been entirely explored.

### 3.1. Breast Cancer

Breast cancer (BC) is responsible for the highest rate of cancer incidence and the second-highest rate of cancer mortality among women [104]. Thus, identifying potential snoRNA biomarkers and therapeutic targets of the disease is urgent. Recent studies have begun to elucidate snoRNAs’ roles in BC. SNORA7B was found to reduce tumorigenesis by reducing the proliferative, invasive, and migratory capabilities of BC cells in knockdown conditions [68]. ACA7B may regulate the acquisition of cancer’s functional hallmarks through the pseudouridylation of 28S rRNA, though the exact mechanism for this remains unknown (Table 1) [105].

As in the previous example, snoRNAs have been thought to contribute to cancer progression as a consequence of a dysregulation of their housekeeping functions as facilitators for the modification of rRNAs. However, some snoRNAs putatively drive cancer progression noncanonically. For example, SNORD50A, which canonically methylates the 2849 and 2864 positions of 28S rRNA, was linked to breast cancer through means that contradicted its canonical localization, function, and binding partners [132]. Specifically, Su et al. reported that SNORD50A and its isomer SNORD50B exhibited oncogenic functions in cell lineages with the p53 wild-type variant (Table 1) [106]. Typically, SNORD50A and SNORD50B disrupt p53 function via the recruitment of GMPS and a ubiquitin ligase, TRIM21, in the cytoplasm [106]. This results in the ubiquitination of GMPS, preventing its translocation to the nucleus [106]. Interestingly, SNORD50A/B deletion restored wild-type p53 function, allowing GMPS to bind with p53 and the deubiquitinating protein USP7, stabilizing p53. However, SNORD50A/B binding and GMPS sequestration result in p53 ubiquitination, inducing p53 nuclear exportation and subsequent proteasomal degradation, leading to a functional loss of p53-induced apoptosis in the cell (Figure 4a) [106]. SNORD50A/B was thus identified as an oncogene through its correlation with the development of BC’s hallmark traits, namely, enhanced proliferation, invasion, and the inhibition of apoptosis, which presented clinically as reduced patient survival [106]. In contrast, SNORD50A/B can also act as a tumor suppressor. Siprashvili et al. recorded the sequestration of the mutant K-Ras oncoprotein in the cytoplasm via direct binding by SNORD50A/B, which significantly reduced the farnesylation of K-Ras by FTase, diminishing tumorigenesis facilitated by the Ras-ERK1/2/MAPK pathway (Figure 4a) [107]. Based on these findings, it is clear that snoRNA can contribute to cancer progression through noncanonical pathways and play both oncogenic and tumor-suppressing roles. Additionally, Zhang et al. identified a negative correlation between the expression of SNORD46—an snoRNA commonly overexpressed in obese patients—in patient blood serum and the quantity of tumor-infiltrating NK cells and CD8^+^ T-cells in breast cancer (Table 1) [108]. Furthermore, an examination of SNORD46-inhibited NK cells found that the cells had gained anti-tumor immunity against triple-negative breast cancer samples [108]. These findings demonstrated that snoRNAs can drive breast cancer progression by suppressing immune activity. Patterson et al. reported that, following biogenesis, SNORD93 was further processed into an snoRNA-derived RNA (sdRNA) that regulated the sarcosine metabolism-related protein Pipox in a functionally similar manner to an miRNA (Figure 4a and Table 1) [110]. The overexpression of sdRNA93 was correlated with increased invasiveness, whereas its inhibition caused a loss of invasive capabilities [110]. Lastly, Su et al. identified that several SNORDs and the C/D box methyltransferase fibrillarin were overexpressed in breast cancer cells and resulted in increased ribosome biogenesis, a recognizable hallmark of several cancers [133,134,135,136]. FBL-overexpressing cells acquired anchorage-independent proliferation and even resistance to doxorubicin in one cell line [137]. FBL was observed to be regulated by p53, which suppresses its nucleolar function through direct binding to elements on intron 1 of the FBL gene, and Myc [137,138]. The downregulation of p53 or a significant loss of function—especially as a characteristic of the mutant phenotype—appeared to prevent this regulation, causing FBL overexpression and modulating FBL activity at known methylation sites, which correlated with increased tumorigenesis and colony-forming capabilities in breast cancer tumors via increased FBL and IGF1R-embedded IRES, a noncanonical initiator of protein translation whose expression is necessary for FBL-induced proliferation [133,137]. The levels of some snoRNAs were modulated in response to p53 suppression [137]. However, critically, the study found no connection between the differential expression of snoRNAs and rRNA methylation levels at known sites, suggesting that snoRNA has a minimal role to play beyond acting as a structural component of the snoRNP that facilitates rRNA methylation by FBL—likely the main player in this mode of oncogenesis [137]. FBL has other well-characterized pathogenic roles involving interactions with classic oncogenes, so its role as a major player in cancer is not unprecedented [139]. Altogether, these findings represent the involvement of a canonical snoRNA pathway protein in cancer development. However, individual snoRNAs do not appear to be the main driver of cancer progression in this mode of oncogenesis. Although additional research is necessary to fully uncover the full scope of the regulatory functions of snoRNA, many recent snoRNA studies point to noncanonical interactions with regulatory proteins as the primary pathogenic mechanisms of cancers involving dysregulated snoRNA as the primary driver of the disease, rather than canonical methylation and pseudouridylation (Figure 4).

### 3.2. Colorectal Cancer

Colorectal cancers (CRCs) account for the second-most cancer deaths in the United States [140]. In many recent studies, the development of cancer and its associated traits have been attributed to the dysregulation of snoRNAs. SNORA21, for example, has been extensively documented in recent years. Yoshida et al. found SNORA21 to be significantly upregulated in CRC cell lines and discovered a cell-cycle modulatory function of the SNORA by observing a decrease in CRISPR–SNORA21-transfected knockdown cells in the S-phase compared to controls, indicating oncogene-like functionality (Table 1) [111]. Yuan et al. described decreased tumor growth following the oncolytic, adenovirus-mediated delivery of SNORD44 and GAS5 into rapamycin-treated cells, proposing SNORD44 as a potential therapeutic solution in CRC (Table 1) [112]. SNORD1C appeared to play a critical role in the Wnt signaling pathway. Liu et al. found a direct correlation between SNORD1C regulation and the expression of β-catenin and its target transcription factor TCF7, in the Wnt signaling pathway (Table 1) [113].

Among the snoRNAs, however, SNORA71A (U71A) appears to play a particularly significant role in CRC due to its regulatory functions in the NF-κB signaling pathway (Table 1). SNORA71A, an antisense, intron-encoded, oncogene-like snoRNA, was significantly overexpressed in CRC cells and caused increased cell proliferation, migration, and invasion while also being implicated in the LPS-induced form of the NF-κB signaling pathway through LBP, a sense-strand encoded, SNORA71A-targeted binding protein to lipopolysaccharide (LPS) [114,141,142]. LPS is most commonly recognized as a normal product of Gram-negative bacteria metabolism, in which it is a major component of the bacteria’s outer membrane and elicits an inflammatory immune response [143]. However, it is also abundant in the intestinal lumen, where the nonpathogenic bacteria produce and secrete the molecule [144]. In regards to its connection with CRC, LPS is essential to the NF-κB pathway, which terminates with the translocation of the NF-κB transcription factor to the nucleus, where it associates with the TNFα promoter to begin gene transcription (Figure 4a) [145]. This suggests a possible pathway through which SNORA71A could influence cancer progression. Furthermore, SNORA71 was previously proven to pseudouridylate uridine 410 (U410) on 18S rRNA [146]. Although the specific effects of each pseudouridine on 18S rRNA are still unclear, the differential expression of snoRNAs—leading to differential pseudouridylation—has been documented to impact ribosomal functional integrity [48]. Penzo et al. described a loss of translational fidelity in dyskerin-depleted cells, resulting in effects including nonsense suppression and amino acid misincorporation [147]. On the other hand, the upregulation of another H/ACA box snoRNA, U19, reportedly caused an increased pseudouridylation of rRNAs, resulting in enlarged global translational efficiency [148]. It is reasonable to speculate that 71A may play a similar role, modulating the rate of transcription of NF-κB proteins in the altered ribosome and inflating the translocation rate of the NF-κB transcription factor to the nucleus, resulting in the increased proliferation, migration, and invasion observed in SNORA71A OE conditions. Alternatively, SNORA71A has been implicated in breast cancer, where it is reported to bind directly to an RNA-binding protein, G3BP1, recruiting it to regulate the stability of ROCK2 mRNA in the TGF-β signaling pathway (Figure 4a) [115]. Such studies seem to suggest that SNORA71A may bind to LBP in a similar manner in CRC, contributing to an increased binding of LBP to its cognate partner protein and dysregulating the NF-κB pathway, causing the cancer’s hallmark characteristics to manifest (Figure 4a). With many components of the NF-κB pathway offering potential as clinical and diagnostic biomarkers in CRC, understanding SNORA71A’s role is vital to determining its viability as a biomarker candidate [149]. Further research using mass spectrometry, RIP-qPCR, Western blotting, and other assays could be the key to determining SNORA71A’s partners, identifying the effect of U410 pseudouridylation on ribosomal function, and describing how U71A affects LBP and its participation in the NF-κB pathway. Crosstalk between β-catenin of the Wnt signaling pathway and NF-κB, which was also shown in CRC cells, likely also impacts downstream elements of NF-κB and must therefore be further investigated to fully characterize SNORA71A [150].

### 3.3. Hepatocellular Carcinoma

Liver cancers were the sixth most diagnosed cancer as of 2020, totaling just over 900,000 new cases and accounting for the third highest mortality of cancers globally, at approximately 830,000 deaths, in the same year [151]. In 2023, these cancers are estimated to comprise 41,210 (2.1%) newly diagnosed cases and 29,380 (4.8%) cancer-related deaths in the US [152,153]. Surprisingly, between 1975 and 2020, both the incidence and mortality rates have remained relatively unchanged, which may be due to the complex host of factors that play a role in the disease [153]. Previous exposure to HBV and HCV, a history of smoking, and alcohol abuse were shown to increase the risk of developing liver cancer [154]. This review will focus on hepatocellular carcinomas (HCCs), which account for 90% of liver cancers and define the majority of current snoRNA research on liver cancer [155].

HCC comprises one of the most extensively researched topics regarding snoRNA in recent years. Xie et al. identified a set of six snoRNAs, including SNORD46, whose dysregulation could serve as a potential marker of poor prognosis in HCC (Table 1) [109]. Xu et al. suggested a pathway by which the suppression of SNORD113-1 as a consequence of a heightened methylation of CpG islands on the promoter region may contribute to tumorigenesis by inhibiting the phosphorylation of ERK1/2 of the MAPK/ERK pathway—in a similar manner to SNORD50A/B in breast cancer—which has been shown to promote cell proliferation and tumor growth in other cancers (Figure 4a and Table 1) [107,116,156]. Wu et al. reported that SNORA11—whose expression was clinically correlated with poorer survival and higher recurrence over 5 years—is commonly overexpressed in HCC and involved in the hyper-phosphorylation of PI3K and AKT, causing the downstream activation of cyclin D1 and augmenting cell proliferation, migration, and invasion (Table 1) [117]. SNORA11 OE xenografts in mice also demonstrated increased tumor growth compared to controls [117]. Wang et al. found that snoRNA-U2-19, upregulated in HCC samples, mediated cell cycle progression and apoptosis via Wnt/β-catenin signaling by facilitating the translocation of β-catenin to the nucleus (Figure 4a and Table 1) [118]. SNORA42 is significantly overexpressed in HCC and suppresses the function of the p53-p21 pathway, which is necessary to induce G1 cell cycle arrest, allowing SNORA42-upregulated HCCs to evade damage control mechanisms and inhibit p53-mediated apoptosis (Table 1) [119,157]. However, the exact mechanism by which SNORA42 regulates p53 is unknown. EMT markers were reduced under SNORA42 knockdown conditions, whereas cell proliferation and the invasion of the surrounding tissues in vitro and tumor growth in vivo were all markedly increased in overexpression conditions [119]. Thus, SNORA42 plays a critical oncogenic role in some HCCs.

A unique study by Fu et al. identified a pathway that implicated the transmembrane cation channel TRPM8 and SNORA55 in oncogenesis (Table 1) [120]. To contextualize these findings, classical models of cancer cell metabolism describe cancer cells reliant on glycolysis rather than oxidative phosphorylation for adenosine triphosphate (ATP) synthesis—likely as an optimization of cellular resource allocation in which glucose molecules normally consumed in cellular respiration are committed to the production of the essential biomolecules necessary for cell proliferation, which is known to place a significantly greater demand on the cell for carbon than ATP synthesis [158]. The complex set of mutations that inhibit cytosolic pyruvate translocation to mitochondria and trigger the overproduction and accretion of lactate, among impacting other cellular mechanisms, collectively produce the classical model of dysfunctional cellular respiration known as the “Warburg Effect” [159]. Additionally, this effect was falsely identified as a compensatory mechanism for failing mitochondrial function [158]. Consequently, mitochondrial activity in cancer has been largely disregarded in the literature until recently. Interestingly, one study found a mode of oncogenesis driven by the SNORA55-dependent regulation of mitochondrial function [120]. TRPM8 was found to be significantly upregulated in HCC tumors, and its expression was positively correlated with tumor abundance and size, while mitochondrial size and relative nucleoli abundance were reduced in murine cells with pharmacologically inhibited TRPM8 [120]. The study noted increased liver activity in TRPM8 knockout (KO) mice, showing reduced blood-serum concentrations of certain biomarkers of liver damage and low-density lipoproteins (LDLs) without any significant alteration to healthy bilirubin and high-density lipoprotein (HDL) levels in the TRPM8 KO murine serum samples compared to wild types [120,160]. These morphological characteristics and lab-obtained data presented in patient-derived and cell line-derived xenografts as slowed cell proliferation and increased apoptosis in TRPM8-knockdown HCC cells compared to controls, but only one cell line demonstrated a marked oncogenic effect as a consequence of overexpression conditions [120]. Furthermore, mitochondrial activity was reduced—reflected by reduced oxygen uptake, cellular ATP concentration, and mitochondrial potential gradient, a necessary condition for oxidative phosphorylation—and mitochondria-dependent apoptosis via BAX was inhibited in TRPM8-expressing cells in response to the introduction and binding of its agonist, WS12 [120,161].

In addition, the expression of SNORA55 was directly related to TRPM8 expression [120]. The H3K4 tri-methylation modification, which was demonstrated to be necessary for normal gene expression, was reduced at sites along the promoter region of SNORA55 in TRPM8 knockdown conditions, which suggests that TRPM8 may regulate SNORA55 through a H3K4-dependent mechanism [120]. A Hi-C assay confirmed the presence of several additional heterochromatin compartments, indicating epigenetic silencing at sites corresponding to the loci of differentially expressed genes as a result of TRPM8 knockdown [120]. Lastly, the authors confirmed that SNORA55 was overexpressed in HCC and regulated key proteins of the oxidative respiratory system [120]. SNORA55 was shown to bind to those proteins, enhancing ATP synthesis, augmenting mitochondrial activity, and effecting the morphological and functional changes observed in HCC (Figure 4d) [120].

Regarding future studies, disrupting the activity of TRPM8 through SNORA55 appears promising. Notably, SNORA55 was present in the nucleus, cytoplasm, and mitochondria of HCC cells, indicating that its function may be as a messenger molecule, in contrast to many other snoRNAs that have highly specific localizations [120]. Perhaps inhibiting SNORA55 motility offers potential as a therapeutic solution for TRPM8-driven oncogenesis in HCCs.

### 3.4. Renal Cell Carcinoma

Renal cell carcinomas (RCCs) are the seventh most commonly occurring cancer and are estimated to comprise approximately 2.4% of cancer deaths in the US in 2023 [153]. Most RCCs are classified as the clear cell, papillary, or chromophobe subtypes [162]. However, clear-cell RCCs account for the vast majority of RCC cases [163]. As reports of snoRNAs implicated in this cancer are relatively scarce, the following summary will be brief.

Zhao et al. constructed independent risk models for survival based on the expression of three snoRNAs for high-risk and low-risk groups, which were demonstrated to have strong correlations with the results of other established pathological and prognostic grading systems in clear-cell RCC [121]. In addition, the authors speculated that SNORA70B, overexpressed in the high-risk signature, or its host gene may contribute to RCC through Wnt signaling, and, as no corresponding snoRNP complex was found, this was hypothesized to be via a noncanonical binding to a protein in the pathway (Figure 4a and Table 1) [121]. Similarly, Shang et al. identified two snoRNAs and Zhang et al. found three snoRNAs whose expressional signatures were effective as diagnostic biomarkers in early-stage clear cell RCC [122,164]. Future studies are necessary to clarify the unexplored mechanisms by which these snoRNAs may contribute to RCC.

### 3.5. Lung Cancer

Despite declining mortality and incidence, lung cancers still rank the highest in cancer lethality worldwide [140,165]. A combination of late stages of diagnosis and disparities in the socioeconomic determinants of health, such as education, consistent employment, and food and housing security, have fueled global LC deaths for decades [166]. Hence, the search for viable targets as well as a stronger understanding of the disease’s pathogenic mechanisms are particularly needed. As few articles have been published on small-cell lung cancer, this section will focus on non-small-cell lung cancers. Here, we summarize snoRNAs’ mechanistic roles in the disease as covered in recent studies.

#### Non-Small-Cell Lung Cancer

Non-Small-Cell Lung Cancer (NSCLC) includes adenocarcinoma, squamous cell carcinoma, and large cell carcinoma and accounts for over half of annual lung cancer cases [167]. Interestingly, snoRNAs with prominent roles in the disease have been identified. For example, SNORA38B localizes in the nucleus, where it binds with the transcription factor E2F1 at several sites, including the critical hydrogen bonding site shared between 38B-U97 and E2F1-Arg165, which was confirmed via an RIP assay and later checked with qRT-PCR and an anti-E2F1 probe in a Western blot analysis (Table 1) [123]. SNORA38B was identified to interact with the GAB2/AKT/mTOR pathway by enhancing the binding affinity between E2F1 and the promoter of the AKT inhibitor GAB2, preventing the phosphorylation of substrates by the master phosphorylator AKT (Figure 4a) [123,168]. The downstream targets of AKT include transcription factors of the FOXO family—proteins with many regulatory functions, including cell cycle regulation and apoptosis—and mTORC1, an oncogene-like protein complex consisting of mTOR, Raptor, mLST8 subunits, PRAS40, and Deptor [169,170,171]. In one pathway, AKT phosphorylates TSC2, inhibiting RHEB, causing the activation of mTORC1, which enlarges the rate of protein synthesis, ultimately contributing to increased cell proliferation [169,171]. In the study, cell proliferation and transwell assays demonstrated increased proliferative and invasive capabilities in vitro, which translated to enhanced tumorigenesis in nude mice [123].

In addition, an analysis of the tumor microenvironment revealed a simultaneous reduction in CD3^+^CD8^+^ T-cell infiltration and inflation in the tumor-resident activity of CD4^+^FOXP3^+^ T-reg cells in SNORA38B OE conditions, while the exact opposite effect was observed in SNORA38B KO samples [123]. SNORA38B OE appeared to coincide with both increased Treg cell activity and increased cellular exportation of an interleukin, IL-10, which was observed to contribute to the former (Figure 4b) [123]. As established in previous studies, IL-10 is a well-documented anti-inflammatory cytokine that regulates immune homeostasis through the inhibition of pro-inflammatory cytokine production in macrophages and lymphocytes [172,173]. Cuesta et al. found that IL-10 was transcribed—independently of IRF-2 interferon regulation—in response to LPS stimulation at LPS receptors, and Riley et al. found IL-10 bound to the extracellular region of the IL-10 receptor, causing the phosphorylation of JAK1 and Tyk2, two kinases bound to the pairs of intracellular α (R1) and β (R2) chains of the IL-10 receptor, respectively [173,174,175]. Activated JAK1 and Tyk2 phosphorylated STAT3—although later studies found that the isoform of STAT targeted was dependent on PI3K signaling in tandem with cytokine-induced gene activation by cytokines such as IFNγ and determined the initiation of anti-inflammatory and pro-inflammatory reactions by STAT3 and STAT1, respectively [175,176]. Following tyrosine phosphorylation, STAT homodimerizes and translocates to the nucleus, where it is recruited by cofactors that are bound to the promoters of IL-10 anti-inflammatory signaling products and help specify binding sites of STAT (Figure 4b) [175,177,178,179]. Interestingly, while the products of IL-10-induced gene activation, notably SOCS1 and SOCS3, appear to contribute to immuno-inflammatory effects, such as triggering the production of TNF-α, IFN-γ, IL-6, and nitric oxide, both isoforms of SOCS demonstrated marked inhibitory characteristics by binding to JAKs, preventing their activation and suppressing immuno-inflammatory gene transcription [178]. IL-10 is secreted by CD4^+^ T cells, cytotoxic T cells, B cells, dendritic cells, and phagocytes, among other immune cells, in vivo [180,181,182,183]. However, as Zhuo et al. confirmed, the SNORA38B-driven overexpression of IL-10—verified using ELISA on culture supernatants—was observed in NSCLC cells, which further supports NSCLC IL-10 secretion as a pathogenic mechanism through which tumors evade the body’s innate, anti-cancer autoimmune responses (Figure 4a) [123]. Altogether, the findings jointly substantiate that SNORA38B is an oncogene that drives cancer by enhancing tumor growth and suppressing cancer-regulating immune activity.

Regarding future research avenues, the mechanism by which SNORA38B regulates IL-10 must be elucidated. Furthermore, the study proposed SNORA38B-locked nucleic acid (LNA) treatments, which induced a knockdown effect on SNORA38B, to prime tumors for further immune checkpoint blockade (αPD-1, αCTLA-4) therapies, which were used on mice and resulted in increased CD3^+^CD8^+^ T-cell infiltration, reduced CD4^+^FOXP3^+^ Tregs infiltration, and vastly decreased tumorigenesis [123]. Combined immunogenetic therapies offer a promising solution to highly immunogenic cancers involving dysregulated snoRNA and should be investigated in the context of other cancers. Additionally, IL-10 is putatively linked to other non-cytokine signaling pathways, including the NF-κB1 (p50) homodimer of the NF-κB pathway, which appears to regulate IL-10 expression by binding to the IL-10 proximal promoter in macrophages [172]. LPS, an upstream protein in the NF-κB pathway, is well known to stimulate the production of cytokines [184,185]. snoRNAs, including SNORA71A, have been documented to regulate the binding partners of LPS [114]. Thus, the possibility of crosstalk between IL-10 in the GAB2/AKT/mTOR pathway and the NF-κB pathway should be studied for the potential co-regulation of IL-10-induced immune activity based on multiple snoRNAs.

### 3.6. Pancreatic Ductal Adenocarcinoma

Pancreatic cancers have experienced significant global increases in both incidence and mortality in recent decades, and current trends are expected to continue into 2040 [186,187]. As a consequence of relatively mild symptoms in the early stages leading to late diagnoses, the disease often results in poor prognosis and a high proportion of cases of metastatic disease [188,189]. Pancreatic ductal adenocarcinomas (PDACs) account for nearly 90% of pancreatic tumors; thus, they will be the focus of this review [190]. snoRNAs’ roles in PDACs remain relatively unexplored. Thus, we will summarize their reported roles in the literature.

Cui et al. found that overexpressed SNORA23 increased invasiveness and enabled anchorage independence in PDAC cells through the downstream regulation of SYNE2, though no specific mechanism was proven (Table 1) [124]. Chen et al. found that SNORD35A promoted the growth of pancreatic stem cells and facilitated the EMT transition, which enhanced the proliferative, migratory, and invasive capacity of PDAC cells (Table 1) [125]. The authors found that SNORD35A was necessary for the activation of the c-Met tyrosine kinase, which bound HGF to induce the self-phosphorylation of its tyrosines [125]. c-Met, previously implicated in many cancers, may then influence cancer development through various established oncogenic signaling pathways [125].

Lastly, Kitagawa et al. profiled early-stage pancreatic tumors based on the serum levels of two exosomal snoRNAs, SNORA74A, SNORA25, and mRNAs, which, in combination with the pre-existing CA19-9 tumor marker, was effective in distinguishing tumors from controls [127]. Moreover, the snoRNAs included in this study were found to bind to IGF2BP3, a transcriptional regulator protein with well-documented roles in various cancers’ pathologies, bound to the ARF6 transcript, an mRNA included in Kitagawa’s final profile, and played a crucial role in the formation of membrane protrusions (Figure 4a, and Table 1) [127,191]. Mechanistically, in PDAC, IGF2BP3 complexes with other mRNAs and enters cytoplasmic stress granules (SGs)—possibly through interactions with the SG marker partner proteins with which it colocalizes—in the perinuclear region of the cytoplasm for cellular trafficking to actin-rich membrane protrusions, also known as lamellipodia (Figure 4a) [191]. At lamellipodia, the IGF2BP3–mRNA complex disassembles, releasing the mRNA for local transcription [191]. Thus, SG-associated IGF2BP3 appears necessary for the proper, specific localization and function of mRNAs, such as ARF6, whose lamellipodial localization, expression, and function were dependent on IGF2BP3 expression, whereas its cytoplasmic-localized counterpart’s expression was not [191]. In the existing lamellipodia of PDAC cells, the IGF2BP3-dependent expression of ARF6 drove the manipulation of actin into new filaments that promoted the formation of more lamellipodia [191]. Furthermore, although the mechanism of ARF6 packaging into exosomes is unknown, it stands that ARF6 was overexpressed in cells in protrusions in the cell membrane at the invasive fronts of PDACs—which are consequently more mobile—relative to the cells of the main tumor body [127,191]. Based on these traits, ARF6 migration to lamellipodia via SGs likely conveys an advantage in motility in peripheral PDAC cells, ultimately providing the tumor with enhanced invasive and metastatic capabilities [191]. Therefore, the snoRNAs’ potential influence on IGF2BP3′s activity as a mediator of mRNA transit to lamellipodia should be investigated for its potential impact on overall cancer aggressiveness and malignancy.

### 3.7. Melanoma

Melanoma accounted for approximately 1.7% of global cancer diagnoses in 2020, with over 324,000 new diagnoses that year [192]. Inadequate UV protection outdoors remains the disease’s most common pathological origin [193]. Although melanoma mortality has decreased in recent years owing largely to the introduction of immune checkpoint and BRAF/MEK kinase combination inhibition therapies as of 2018 [194], melanoma incidence has risen precipitously, particularly in regions with predominantly light-skinned populations, and survival with late-stage disease remains low [192,193,194,195]. In addition, nearly ⅓ of patients present with metastatic disease, with around a 50% rate of primary non-response to checkpoint therapies and a comparable rate of initial responders with metastatic disease developing secondary resistance [196,197]. Combined with projected increases in diagnoses, the disease’s global burden is expected to rise in future years [198].

Melanomas arise in melanocytes, a neural crest-derived cell responsible for the synthesis of the pigments, eumelanin and pheomelanin, which are delivered in melanosomes by Langerhans’ dendrites to neighboring keratinocytes and oriented around nuclei, where they can form supranuclear caps to protect DNA by absorbing ionizing wavelength radiation and scavenging reactive oxygen species [199,200,201]. Melanomas are typically categorized into major subtypes by either the primary tumor’s tissue of origin or its growth pattern. The former classification consists of four major subtypes which correspond to groups of tumors with similar mutational profiles: cutaneous, the most common type, which occurs in non-glabrous skin; acral, which occurs primarily in glabrous skin of the palms and the soles of the feet; mucosal, the rarest, which occurs on melanocytes of the mucosal lining of internal tissues; and ocular melanomas, which occur in both uveal and retinal melanocytes [202]. The latter classification consists of superficial spreading melanomas, which comprise the vast majority of melanomas, followed by the nodular, lentigo, and acral lentiginous subtypes [203].

snoRNAs’ roles in melanomas are underexplored, with existing snoRNA-focused research on the disease largely involving the construction of expressional signatures for prognostic uses. Consequently, literature reports of the underlying molecular mechanisms are highly limited. Wang et al. constructed a prognostic model based on the expressional signature of 12 snoRNAs whose risk score was positively correlated with the presence of naive B-cells and CD4^+^ T-cells in the tumor microenvironment and had potential for predicting patient sensitivity to immune checkpoint inhibition therapies (Table 1) [128]. Yi et al. created a prognostic model using four other snoRNAs, which may be associated with CNS activity [204]. Rahman et al. found that SNORA24 was downregulated in cutaneous melanoma cells, though a specific oncogenic mechanism was not identified [205]. Lastly, Lunavat et al. found SNORD83A and SNORD89, among other small RNAs, accumulated at greater concentrations in exosomes compared to host cells, with SNORD89 particularly enriched in apoptotic bodies, which suggests a potential role for these snoRNAs in cell-to-cell communication and the transport of molecules to recipient cells (Figure 4c and Table 1) [129]. SNORD83A canonically guides the 2′O-methylation of 18S rRNA, while SNORD89 canonically guides the methylation of Bim and was found to epigenetically downregulate this gene in endometrial cancer [206,207]. Future studies are needed to clarify the unknown molecular interactions of these snoRNAs in melanoma.

### 3.8. Leukemia

The development of leukemia putatively occurs as a consequence of mutations in pre-hematopoietic progenitor cells, such as the hemangioblast; hematopoietic progenitor cells, such as hematopoietic stem cells, multipotent stem cells, and oligo-lineage progenitors; and fully differentiated blood cells [208]. More functionally specialized cells likely require a greater number of mutations in key genes to reacquire their self-renewing capabilities and inhibit apoptosis [208]. The rate of differentiation and the self-renewing capacity of stem cells are influenced by many factors, such as cytokine-feedback regulation, which is influenced by blast population size and determines the number of replications a cell can undergo before it reaches terminal differentiation—as demonstrated by Mangel et al.’s differential model [209,210]. Leukemic stem cells can be distinguished from healthy stem cells by their cell-surface markers, and the disease is generally divided into four major subtypes: the chronic and acute variants of myelogenous and lymphocytic leukemias [208,211].

Leukemia comprises approximately 46.6% of cancer diagnoses in children and teens in the 1–19 age range in the US [212]. Globally, heightened incidence and mortality rates have been attributed to lower averages in socioeconomic status and insufficient access to modern healthcare, making the disease the primary focus of pediatric cancer research [213,214]. Owing to improved early diagnosis and treatment options, the disease has seen worldwide reductions in age-standardized incidence and mortality rates in recent decades [215]. Leukemia ranked 10th in overall cancer incidence in the US as of 2021 [216]. However, considering its prevalence among youth populations, further research is necessary to minimize the disease’s burden on this demographic [216]. Naturally, the functions of snoRNA have been scrutinized in the context of leukemogenesis and the progression of the disease.

For example, Pauli et. al. directly correlated SNORD42A KO with a loss of colony-forming capabilities and the slowing of cell proliferation and SNORD42A deletion with a reduction in cell size in acute myeloid leukemia (AML) [130]. SNORD42A canonically mediates 2′O-methylation at U116 of 18S rRNA, and SNORD42A-guided methylation may alter ribosomes’ morphological structure, causing the observed reduction in the translation of ribosomal proteins (Figure 4d) [130]. Considering that SNORD42A KO conditions resulted in a decrease in overall translational activity, the authors correlated the modulated cell functions in AML with a loss of the U116 modification (Table 1) [130]. However, given the broad spectrum of snoRNAs identified to participate in cancer regulatory networks through noncanonical protein interactions, further studies must study noncanonical binding partners for SNORD42A in order to fully characterize its role in leukemia and provide a more holistic view of this snoRNA’s functions in the cell. As SNORD42A binds to FMRP and nucleophosmin 1, these proteins may be strong candidates to be investigated for an alternative mode of leukemogenesis [130].

In another study, Zhou et al. identified a novel function for the chimeric oncoprotein AML1-ETO via the activation of AES, as a regulator of the 2′O-methylation activity of several C/D box snoRNAs, which was essential to leukemogenesis [126]. AES deletion diminished clonogenicity in vitro and stunted engraftment in secondary and subsequent transplantations of murine fetal liver cells—which are important to HSC differentiation and replication during embryonic development—in mice [126,217]. AES deletion additionally resulted in a reduction in c-Kit surface markers, a common receptor whose expression is directly correlated with lower survival and complete remission rate in AML, and caused a reduction in translational efficiency observed via a puromycin inhibition assay [126,218]. In a clinical trial, AML patients who overexpressed many of a large cohort of snoRNAs showed weaker responses to induction chemotherapy [126]. Some C/D box snoRNAs’ activities were directly related to AES expression in AML1-ETO-positive cells [126]. For example, AES depletion impaired SNORD43 and SNORD32A 2′O methylation activity, while the expression of canonical C/D box snoRNP proteins was unaffected, indicating the direct regulation of snoRNA expression (Figure 4d and Table 1) [126]. AES was found to mediate snoRNP activity through an RNA helicase, DDX21, whose binding effectiveness was regulated by AES, and knockdowns of several snoRNAs suppressed by AES, including SNORD43, resulted in reduced clonogenicity, cell size—and, by implication, ribosome biogenesis—and the protein synthesis rate (Table 1) [126]. Notably, the knockdown of SNORD14D, also suppressed by AES, reduced colony formation without affecting methylation levels, suggesting a possible noncanonical method of oncogenesis (Table 1) [126].

### 3.9. Lymphoma

Lymphomas consist of an immunophenotypically diverse set of cancers of lymphoid progenitor cells and fully differentiated lymphocytes, including B-lymphocytes and, much less commonly, T and NK cells, that primarily originate in the lymphoid organs but can occur extranodally, impacting many systems [219]. Consequently, the different subtypes of the condition range in severity and can present with vastly different clinical traits [219]. Although the disease possesses a great degree of heterogeneity, lymphomas are classically divided into the non-Hodgkin’s and Hodgkin’s disease groups, which can be split into additional subdivisions based on prognosis and other unifying immunophenotypic and histological features [219,220,221]. Hodgkin lymphomas, for example, are distinguished by the presence of Reed–Sternberg cells, a rare, morphologically distinct cell type that can arise as a consequence of severe nonsense mutations that occur in place of the normal, nonpathogenic hypermutation of B-cells during B-cell maturation in germinal centers and can lead to the acquisition of the cell’s characteristic duplicated nuclei and the suppression of typical B-cell receptors, among other traits, possibly including the inhibition of apoptosis [220,222,223,224]. Some subtypes of Hodgkin’s and non-Hodgkin’s lymphoma can be even further categorized into smaller, yet genetically unique, classes [225]. Despite ranking 15th in incidence globally, lymphoma is the most commonly diagnosed cancer in the 15–19 age range and predominantly affects the 15–35 and 55+ age ranges in a bimodal distribution [226]. Non-Hodgkin’s lymphomas have been estimated to comprise a large majority of lymphoma cases and approximately 3.3% of cancer-related deaths, whereas Hodgkin’s lymphomas consist of a small fraction of overall lymphoma cases [153]. Epidemiologic trends also appeared to have some correlation with socioeconomic disparity, with a higher incidence and lower mortality in high-income areas and the opposite pattern in low-income regions, likely influenced by diagnostic capabilities and treatment availability [221,227]. Compared to other cancers, the known mechanistic roles for snoRNA in lymphoma are relatively lacking. With some notable exceptions to current research trends, such as Li et al.’s risk model relating SNORD1A, SNORA60, and SNORA66 expression to patient survival over 1, 3, and 5 years (Table 1), most recent mechanistic studies have instead reported the involvement of SNHGs in the disease [131]. Although SNHGs’ exons processed into lncSNHGs function differently and localize independently of their intron-embedded snoRNAs, the similar mutation and dysregulation signatures, or patterns, present on both snoRNAs and their coupled lncSNHGs may also influence both RNA species’ correlations to clinicopathological traits and patient outcomes in cancer [9,66]. However, it must be noted that snoRNAs and their host genes’ expressions can have little to no connection—likely due to nonsense-mediated decay and dual-transcription initiation [67,70]. Still, analyzing SNHGs may provide valuable insights into the roles of their housed snoRNAs in the context of snoRNA-based cancer studies. A brief summary of SNHGs reported in lymphoma will be provided below.

Zhu et al. found that lncSNHG16, upregulated in diffuse large B cell lymphoma (DLBCL), increased cell proliferation and suppressed apoptosis in DLBCL by sponging, or competitively inhibiting, the 5p strand of MIR497, which disrupted miRNA binding at the 3′UTR region of PIM1, a receptor with a putative regulatory role in cell cycle progression, preventing G0/G1 cell cycle arrest and apoptosis in lymphoma cell lines [228]. Similarly, lncSNHG14, identified to be overexpressed in DLBCLs, was reported to be an miRNA sponge of the 3p strand of MIR5590 that upregulated ZEB1, a protein that promotes cell proliferation and the EMT and a transcription factor for SNHG14 and PD-L1, a checkpoint protein which impairs CD8^+^ T-cells’ anti-tumor immune activity [229]. Lastly, SNHG12 overexpression was directly correlated with Ki67 and Pgp expression—a marker of cell division and a membrane transport protein responsible for the cellular export of cancer drugs in cancer cells, respectively—which increased cell proliferation and imparted drug resistance in NK/T-cell lymphomas [230,231]. The snoRNAs encoded within these host genes are the SNORD1 isoforms for *SNHG16*, 30 snoRNAs for *SNHG14*, and SNORA16A, SNORA44, SNORA61, and SNORD99 for *SNHG12* [67,232,233].

Interestingly, a chromosome translocation (1q25:3q27) was recorded to commonly occur in B-cell lymphomas in the *GAS5* gene, whose introns contain SNORD76 [234]. Although the study attributed the cause of lymphomagenesis to the dysregulation of BCL6 in the GAS5-BCL6 chimeric oncoprotein, as the breakpoint of the translocation occurred in the sequence of SNORD76 [234], the consequences of lost SNORD76 function should be investigated to determine whether U76 plays a role in lymphoma. Otherwise, with many of snoRNAs’ roles in lymphoma being unknown, there is great potential for novel studies to further elucidate the field.

## 4. Conclusions and the Road Ahead for snoRNA

snoRNAs, as underexplored drivers of cancer progression, have grown tremendously as a focus of research in recent years. With further reports, the increasingly apparent mischaracterization of snoRNA function has gradually shifted the scientific consensus of snoRNAs from their roles as simple housekeeping transcripts to a functionally diverse class of noncoding RNAs that are directly involved in oncogenesis and are highly effective as diagnostic and prognostic tools. This review aimed to reveal the complex molecular mechanisms of snoRNAs in oncogenesis and identify research gaps where future clarifying studies are necessary.

Broadly, dysregulated snoRNAs mainly drive oncogenesis by binding with proteins to facilitate or suppress the activity of their bound targets or to induce a cascade of downstream malfunctions in a protein’s pathway. snoRNAs can achieve these effects either directly or by facilitating chemical modifications to modulate their target’s motility, stability, and binding affinity, among other characteristics, which effects the mislocalization and dysfunction of downstream proteins. This snoRNA activity impacts proteins involved in the upkeep of normal cellular functions in a variety of cellular localizations, including the nucleus, cytosol, mitochondria, and cell membrane, and thus conveys a diverse range of selective advantages to cancer cells.

Moreover, in the absence of differential regulation, snoRNAs occasionally retain their role in oncogenesis as vital structural elements whose antisense complementarity provides highly specific targeting when assembled into snoRNPs, from which dysregulated canonical snoRNA-binding proteins can cause a loss of ribosomal translational fidelity through modulated rates of 2′O-methylation and pseudouridylation at rRNA sites. Although other mechanisms may exist, noncanonical transcript binding and the disruption of snoRNP activity appear to be the most common. Interestingly, the vast majority of recent studies implicate the former as the primary mode of snoRNA-driven oncogenesis.

Regardless, these processes cause the deleterious malfunction of cellular functions, including but not limited to stem cell differentiation, transcriptional fidelity, ribosome biogenesis, epigenetic regulation, and immunoregulation. snoRNAs’ differential expressions ultimately lead to the acquisition of cancer’s hallmark traits, inducing the EMT and lamellipodia formation as well as enhanced proliferation, invasiveness, and metastasis, which are the pathological hallmarks of heightened lethality in cancers. Overall, these studies characterize snoRNAs not as idle spectators but as a heterogenous group of biomolecules deeply ingrained in the development and progression of cancers.

From discoveries of their involvement in various biological processes and dysregulation in diseases, snoRNAs have grown in clinical relevance too. Specific snoRNA detection could enhance diagnosis and prognosis in early cancer. Dysregulated snoRNA expression profiles in various cancers have proven their efficacy as diagnostic and prognostic biomarkers. Altered snoRNA expression patterns detected in circulating biofluids offer minimally invasive options for cancer detection and monitoring. snoRNAs whose expression signatures correlate with poor clinical outcomes like tumor progression, metastasis, and reduced survival rates, integrated into existing prognostic models, may enhance risk stratification and treatment decision making.

While more work is needed to fully understand the role of snoRNAs in tumors, it is promising for the field that many snoRNA studies, particularly those concerning the construction of expression signatures, lack mechanistic studies, leaving the acting molecular mechanisms of the snoRNAs implicated poorly understood. These reports would be necessary to inform researchers of their value as therapeutic targets, which remains questionable.

Despite this deficiency, targeting dysregulated snoRNAs in cancer treatment presents great therapeutic potential. Strategies using antisense oligonucleotides, small interfering RNAs (siRNAs), locked nucleic acids (LNAs), and CRISPR-based approaches can modulate snoRNA expression, disrupting cancer pathways and inhibiting tumor growth. SnoRNA-based therapeutics offer a new direction for precision medicine. Additionally, snoRNAs are linked to drug resistance in cancer cells, potentially influencing apoptosis, DNA repair, and drug efflux. Understanding snoRNAs’ roles in drug resistance may enable the development of combination therapies that overcome this treatment challenge.

As previously mentioned, snoRNAs are involved in immunoregulation. In cancer, snoRNAs can regulate cytokine and potentially chemokine expression, shaping the tumor microenvironment and influencing immune cell recruitment and activation within tumors, as demonstrated in murine models by Zhuo et al. [123]. Additionally, snoRNAs may modulate the expression or activity of key checkpoint molecules like PD-1, PD-L1, and CTLA-4, desensitizing recipients of checkpoint blockade therapy to immune checkpoint inhibitors (ICIs). Moreover, snoRNAs may interact with other noncoding RNAs involved in immunoregulation, such as miRNAs and lncRNAs, forming complex regulatory networks. Although the precise roles of snoRNAs in dictating immunotherapy responses are still being explored, their immunoregulatory capabilities suggest they could be valuable therapeutic targets for enhancing immunotherapy responses in cancer patients.

Lastly, Dubey et al. published a comprehensive review on the roles of dysregulated snoRNAs in the pathogenesis and treatment of autoimmune disorders [235]. Altered expression levels of snoRNAs have been reported in various autoimmune diseases, including systemic lupus erythematosus, rheumatoid arthritis, and multiple sclerosis. Genome-wide expression studies have identified profiles of differentially expressed snoRNAs in autoimmune disease patients compared to healthy controls, suggesting the role of snoRNAs in autoimmune disease pathogenesis. As snoRNAs can regulate the expression of genes involved in inflammation, cytokine production, and immune cell recruitment, the aberrant snoRNA-mediated regulation of inflammatory pathways may exacerbate autoimmune responses and tissue damage. Evidently, the pathogenic roles of snoRNAs extend far beyond the realm of cancer.

It is our sincere hope as well as the reflection of the current literature that snoRNAs may pave the way for breakthroughs—that through snoRNAs, our efforts to untangle the long-unsolvable network of defective molecular mechanisms within cancer, among other diseases, may be greatly rewarded.

## Figures and Tables

**Figure 1 ijms-25-02923-f001:**
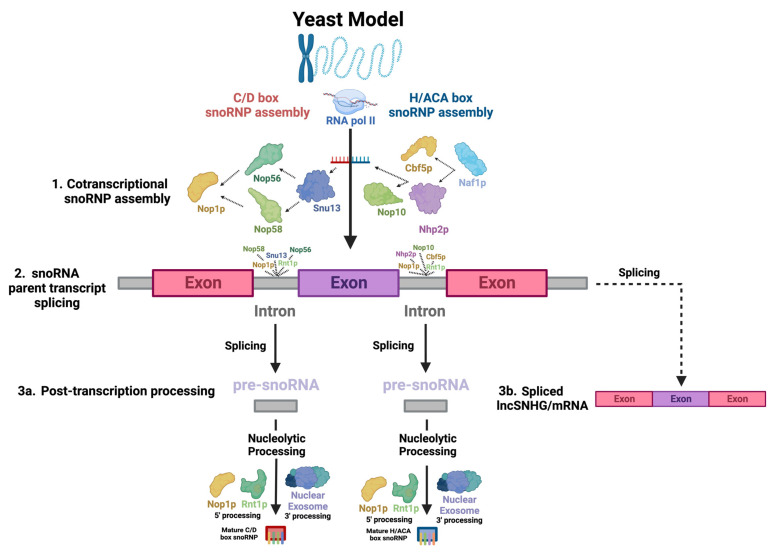
Key steps in snoRNA biogenesis. There appears to be significant conservation between yeast and human snoRNAs, so biogenesis across species is assumed to be comparable. snoRNAs undergo highly complex biogenesis. The diagram depicts the processing of an RNAPII-transcribed, monocistronic, intron-embedded snoRNA, the type of snoRNA most common in humans, into a functional snoRNP. This process begins with the (**1**) co-transcriptional binding of snoRNP core proteins to the snoRNA transcript, which serves as the structural backbone of the complex. This also protects the snoRNA from post-transcriptional nucleolytic degradation. The direction of the arrows indicates the sequence in which interactions between the molecules occur. (**2**) Next, the primary snoRNA “parent” transcript is spliced, releasing snoRNA-embedded introns—the “pre-snoRNAs.” (**3a**) Lastly, mature snoRNAs and associated proteins are excised from introns through nucleolytic processing at the 5′ and 3′ ends, involving Nop1p and Rnt1p and the nuclear exosome, respectively. (**3b**) Exons are spliced into a functional RNA—either an mRNA or lncSNHG.

**Figure 2 ijms-25-02923-f002:**
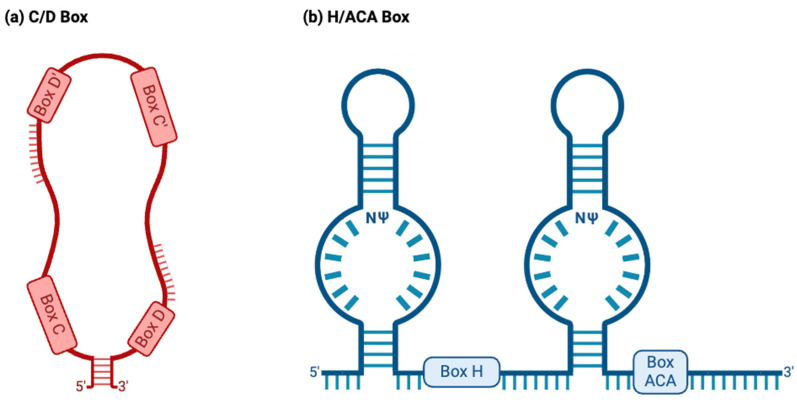
Two major snoRNA families. Typically, snoRNAs are classified into the C/D box or H/ACA box families based on their structure and function. (**a**) C/D box snoRNAs normally have a stem–bulge–stem structure, with C and D box motifs and C′ and D′ box motifs present on opposing ends of the C/D box snoRNA. The C and D boxes, located at the 5′ and 3′ ends, represent the sequence motifs RUGAUGA (R represents a purine base) and CUGA, respectively. C/D box snoRNAs canonically guide the 2′O-methylation of target RNAs, binding to targets using antisense elements located downstream of the D and D′ box sequence motifs (depicted above). (**b**) H/ACA box snoRNAs normally have a hairpin–hinge–hairpin–tail structure, with the H box present on the hinge and the ACA box located on the tail. The H and ACA boxes represent the motifs ANANNA and ACA. H/ACA box snoRNAs also possess complimentary guide sequences. However, these appear within “pseudouridylation pockets”, in which the modification occurs. NΨ: symbol for a pseudouridine nucleotide, representing the location on H/ACA box snoRNAs where uridines are pseudouridylated.

**Figure 3 ijms-25-02923-f003:**
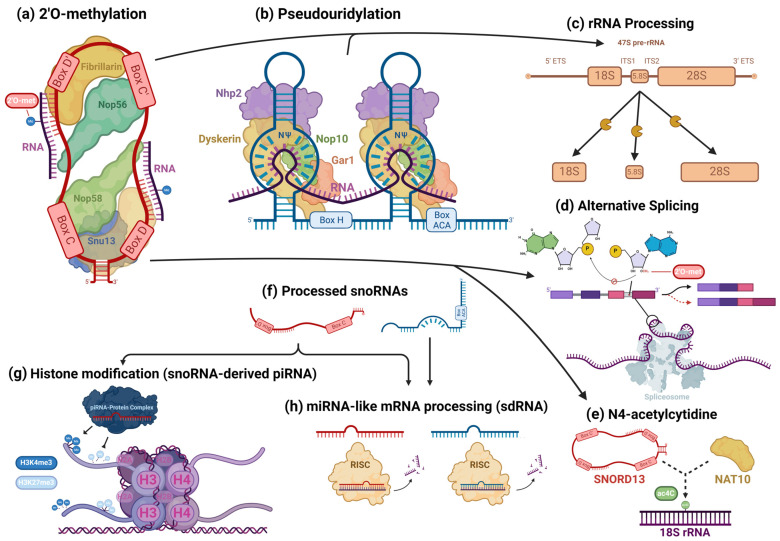
Cellular processes facilitated by snoRNAs. snoRNAs are mostly associated with their role as guides for 2′O-methylation (**a**) and pseudouridylation (**b**) modifications, in which they bind to target transcripts and, using their scaffold-like structure, coordinate snoRNP core proteins that catalyze these modifications. However, snoRNAs are also involved in the epigenetic regulation of a wide variety of targets. In addition to their roles in rRNA processing (**c**), snoRNAs also regulate alternative splicing (**d**) and can guide rarer nucleotide modifications, such as the N4-acetylcytidine modification (**e**). Some snoRNAs are processed into smaller fragments (**f**) and form complexes with piRNA- and miRNA-associated proteins, playing the role of those RNAs. snoRNA-derived piRNAs and miRNAs have been found to regulate histone modifications (**g**) and cleave target RNAs (**h**), respectively, modulating gene expression.

**Figure 4 ijms-25-02923-f004:**
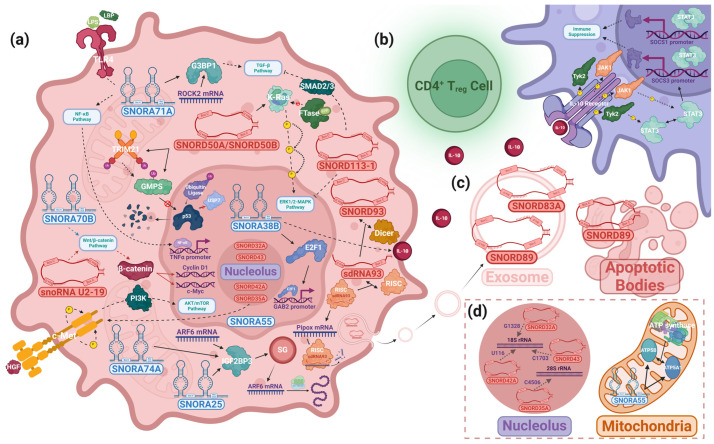
snoRNAs’ pathways of oncogenesis. Mechanistic studies of snoRNA-driven oncogenesis reveal an unexplored realm of interactions between snoRNAs and well-documented cancer pathways. snoRNAs regulate these pathways through a diverse range of proteins. These interactions—which take place in the (**a**) cytosol, nucleus, and (**d**) other organelles, including the nucleolus and mitochondria, and involve intercellular communication via (**b**) cytokines and (**c**) extracellular vesicles—play a vital role in various cancers’ malignant transformations. The above figure depicts several snoRNAs whose mechanistic involvement in oncogenesis is known. The snoRNAs that interact with the same pathway may not necessarily do so by the same mechanisms. Solid arrow: binding interactions. Dashed arrow: the interaction of a molecule that influences its target either by facilitating the target’s modification or through downstream signaling. Tapered arrow: a molecule’s movement. Line ending in dot: the product of the interaction linked to this symbol. Colored arrow (red/blue): the activity of the (red) C/D box snoRNA or (blue) H/ACA box snoRNA within its associated pathway. “No” symbol: indicates an interaction that does not occur in cancer conditions.

**Table 1 ijms-25-02923-t001:** snoRNA interactions in cancer.

snoRNA	Function	Cancer	Targets + (Pathway of Oncogenesis)	Molecular Interactions	Associated Malignant Transformations ^†^	Refs
SNORD50A/B	ONC|TS	BC	GMPS, TRIM1, p53 (p53)|K-Ras (Ras-ERK1/2/MAPK)	Recruits/sequesters GMPS via TRIM1|Recruits/Impairs K-Ras-FTase interaction	Proliferation (↑); Migration (↑); Invasion (↑); Cell Cycle Arrest (↓)|Tumor Growth (↓)	[106,107]
SNORA7B	ONC	BC	Unknown	Unknown	Proliferation (↑); Migration (↑); Invasion (↑); Tumor Growth (↑); Apoptosis (↓)	[68]
SNORD46	ONC|ONC	BC|HCC	Unknown|Unknown (Notch) ^§^	Unknown|Unknown	NK Tumor Infiltration (↓)| Neutrophil + NK Tumor Infiltration (↓)	[108,109]
SNORD93 (sdRNA93)	ONC	BC	Pipox (Sarcosine Metabolism)	Facilitates Pipox 3′UTR cleavage	Invasion (↑)	[110]
SNORA21	ONC	CRC	Unknown (Wnt) ^§^, Unknown (Hippo) ^§^	Unknown	Proliferation (↑); Tumor Growth (↑)	[111]
SNORD44	TS	CRC	Procaspase 3/8/9 (Caspase-Dependent Apoptosis)	Unknown	Proliferation (↓); Tumor Growth (↓); Apoptosis (↑)	[112]
SNORD1C	ONC	CRC	β-catenin (Wnt/β-catenin)	Unknown	Proliferation (↑); Migration (↑); Invasion (↑); Apoptosis (↓)	[113]
SNORA71A	ONC|ONC	CRC|BC	LBP (NF-κB)|G3BP1 (TGF-β)	Unknown—LBP ^§^|Binds G3BP1, stabilizes ROCK2 mRNA	Proliferation (↑); Migration (↑); Invasion (↑)|Proliferation (↑); Migration (↑); Invasion (↑); Tumor Growth (↑)	[114,115]
SNORD113-1	TS	HCC	ERK1/2 (MAPK/ERK)	Inhibits ERK1/2 phosphorylation	Proliferation (↓); Tumor Growth (↓); Apoptosis (↑)	[116]
SNORA11	ONC	HCC	PI3K, AKT (PI3K/AKT)	Unknown—Involved in hyper-phosphorylation of PI3K, AKT	Proliferation (↑); Migration (↑); Invasion (↑); Cell Cycle Arrest (↓)	[117]
snoRNA U2-19	ONC	HCC	β-catenin (Wnt/β-catenin)	Mediates β-catenin nuclear translocation	Proliferation (↑); Tumor Growth (↓); Cell Cycle Arrest (↓); Apoptosis (↓)	[118]
SNORA42	ONC	HCC	p21, p53 (p53)	Unknown	Proliferation (↑); Migration (↑); Invasion (↑); Tumor Growth (↑); Cell Cycle Progression (↑); Apoptosis (↓)	[119]
SNORA55	ONC	HCC	ATP5A1, ATP5B (OXPHOS)	Migrates to mitochondria, binds ATP5A1 + ATP5B	Proliferation (↑); Tumor Growth (↑); Apoptosis (↓)	[120]
SNORA70B *	ONC	RCC	Unknown (Wnt)	Unknown	Nuclei Regularity (↓); Nucleoli Size (↑); Migration (↑); Invasion (↑); Hemoglobin (↓)	[121]
SNORD15A *	ONC	RCC	Unknown	Unknown	Tumor Growth (↑)	[122]
SNORD35B *	ONC	RCC	Unknown	Unknown	Tumor Growth(↑); Migration(↑)	[122]
SNORA38B	ONC	NSCLC	E2F1 (GAB2/AKT/mTOR)	Binds/enhances binding affinity of E2F1 to GAB2 promoter	Proliferation (↑); Migration (↑); Invasion (↑); Tumor Growth (↑); Apoptosis (↓); Anti-Inflammatory Cytokine Secretion (↑)	[123]
SNORA23	ONC	PDAC	SYNE2 (Invadopodia Formation)	Unknown	Invasion (↑); Anchorage-Independence (↑)	[124]
SNORD35A	ONC|ONC	PDAC; LEUK	Unknown (HGF/C-Met)|28S rRNA C4506 (Ribosome Biogenesis)	Unknown|U35 guides snoRNP-driven 28S rRNA C4506 methylation	Proliferation (↑); Migration (↑); Invasion (↑); Tumor Growth (↑)|Proliferation (↑); Translation Rate (↑); Cell Size (↑)	[125,126]
SNORA74A * /SNORA25 *	ONC	PDAC	IGF2BP3 (Stress Granule Assembly)	Binds to IGF2BP3	Lamellipodia Formation (↑)	[127]
SNORD9 * /SNORD14A * /SNORD14E * /SNORD83A * /SNORA5A * /SNORA14A * /SNORA31 * /SNORA65 * /SNORA75 *	ONC	MEL	Unknown	Unknown	Local Cytotoxic Immune Cell Activity (↓)	[128]
SNORD83A/SNORD89	ONC	MEL	Unknown	Accumulates in extracellular vesicles	Unknown	[129]
SNORD42A	ONC	LEUK	18S rRNA U116 (Ribosome Biogenesis) ^§^	U42A guides snoRNP-driven 18S rRNA U116 methylation ^§^	Proliferation (↑)	[130]
SNORD43	ONC	LEUK	18S rRNA C1703 (Ribosome Biogenesis)	U43 guides snoRNP-driven 18S rRNA C1703 methylation	Proliferation (↑); Translation Rate (↑); Cell Size (↑)	[126]
SNORD32A	ONC	LEUK	18S rRNA G1328 (Ribosome Biogenesis)	U32A guides snoRNP-driven 18S rRNA G1328 methylation	Translation Rate (↑); Cell Size (↑)	[126]
SNORD14D	ONC	LEUK	Unknown	Unknown	Proliferation (↑); Translation Rate (↑); Cell Size (↑)	[126]
SNORD1A * /SNORA60 * /SNORA66 *	ONC	LYMPH	Unknown	Unknown	Unknown	[131]

ONC; oncogene, TS; tumor suppressor, BC; breast cancer, CRC; colorectal cancer, HCC; hepatocellular carcinoma, RCC; renal cell carcinoma, NSCLC; non-small cell lung cancer, PDAC; pancreatic ductal adenocarcinoma, MEL; melanoma, LEUK; leukemia, LYMPH; lymphoma. ^†^ “Associated Malignant Transformations” refers to transformations that occur as a result of the snoRNAs’ non-experimentally altered activity in their respective cancers. ^§^ indicates a speculated connection to the corresponding snoRNA. * a snoRNA studied for biomarker viability instead of mechanistic involvement in cancer. ↑ indicates that the transformed cellular function or characteristic is enhanced in cancer compared to healthy conditions. ↓ indicates that the transformed cellular function or characteristic is attenuated in cancer compared to healthy conditions.

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
