# Peer review of "Subverting the Canon: Novel Cancer-Promoting Functions and Mechanisms for snoRNAs"

_ijms, 2024, doi:10.3390/ijms25052923_

Round 1

Reviewer 1 Report

Comments and Suggestions for Authors

I found this as a solid work and its quality is good, however, the author still needs to solve the following problems.

1-      Provide a table of abbreviations with their expansions.

2-      Numerous studies have examined the role of snoRNA in different types of cancer. What benefits does this manuscript have over other research?

3-      What would be possible impact of genetic variants of snoRNA in different cancers and why?

4-      I think the author could discuss the role of snoRNA in immunotherapeutic response of a cancer patient.

5-      Add a discussion on clinical relevance of noncoding snoRNA.

6-      Description of the in vitro data should be included in the paper, it very important in the medical-related paper.

7-      The biggest weakness of the manuscript is the lack of the clinical data and should be included in the medical-related manuscript.

8. The manuscript is based on implications of snoRNA in cancer and oncogenesis. I suggest citing of the following article.

Cancer represents one of the most frequent causes of death in the world. The current therapeutic options, including radiation therapy and chemotherapy, have various adverse effects on patients’ health. E

Hesperidin, a Bioflavonoid in Cancer Therapy: A Review for a Mechanism of Action through the Modulation of Cell Signaling Pathways. https://doi.org/10.3390/molecules28135152.

Comments on the Quality of English Language

Double check the manuscript for grammatical errors.

Reviewer 2 Report

Comments and Suggestions for Authors

The manuscript by Huo al cols is a comprehensive revision of the role of nucleolar snoRNAs in carcinogenesis. Given the length of the manuscript and the  exhaustive treatment of the topic it is difficult for this reviewer to provide useful comments, but I’ll give a couple of hints with the aim not to improve it (a difficult task) but to make it more compelling to readers interested in pathologies other than cancer, even if this means reducing the length of the cancer section.

- It is known that snoRNAs are related with the onset and development of autoimmune diseases. Could the authors also discuss this topic?

- Is there any relationship among snoRNAs and the regulation of alternative/cryptic splicing?

- Authors should deep on the miRNA-like role of snoRNAs. Are these restricted to 3’UTRs? Are these affected by 3’UTR dynamics?

-Figure 4 is very difficult to understand, mainly because of the size of lettering. I would suggest splitting it into different sub-figures with a higher magnification
